# Social determinants of vaccine hesitancy among the Lebanese parents: A cross-sectional Study

Reem Awad[ID]*, Mohamad Rahal[1], Anna-Maria Henieneh[2,3], Pascale Salameh[ID][2,3,4], Robin Farah[5], Michele Cherfane[3,5], Reham Kotb[6], Diana Nakhoul[2], Rêve Khadaj[2], Nayla Habib[7], Katia Iskandar[1,5,8,9]

**1** Department of pharmaceutical sciences, School of Pharmacy, Lebanese International University, Beirut, Lebanon, **2** School of Pharmacy, Lebanese University, Beirut, Lebanon, **3** INSPECT-LB (Institut National de Santé Publique, d'Épidémiologie Clinique et de Toxicologie-Liban), Beirut, Lebanon, **4** Department of Primary Care and Population Health, University of Nicosia Medical School, Nicosia, Cyprus, **5** Gilbert and Rose-Marie Chagoury School of Medicine, Lebanese American University, Byblos, Lebanon, **6** Environmental and Public Health Department, College of Health Sciences, Abu Dhabi University, Abu Dhabi, United Arab Emirates, **7** Lebanese order of Pharmacy, Beirut, Lebanon, **8** Center for Collaborative Research Initiatives in Public Health, Higher Institute of Public Health, Saint Joseph University of Beirut, Riad El Solh, Beirut, Lebanon, **9** Faculty of Public Health, Section 2, Lebanese University, Fanar, Lebanon

* reem.awad.2010@hotmail.com

## Abstract

### Background

Vaccines are vital in preventing infectious diseases and saving millions of lives globally. However, the rise of vaccine hesitancy has hindered adequate vaccination coverage, particularly among children. This study explores the social determinants influencing childhood vaccine hesitancy among a sample of Lebanese parents, focusing on how individuals are born, grow, live, and work.

### Methods

A cross-sectional study was conducted using a self-administered questionnaire that gathered socio-demographic characteristics, factors influencing childhood vaccination, and social determinants of health (SDOH). Vaccine hesitancy was measured using a 23-item scale with two subscales (hesitancy and acceptance/trust) rated on a 5-point Likert scale. SDOH included socioeconomic status), Vaccine Health Literacy, Daily Spiritual Experience Scale, and Discrimination in Medical Settings Scale. Data were analyzed using bivariate tests and multivariable linear regression with vaccine hesitancy as the dependent variable.

### Results

A total of 251 participants completed the survey. The mean parental age was 34.94 ± 8.69 years, with 55.8% being female, and the mean child age was 21.35 ± 11.53 months (range: 0–36 months). Higher vaccine hesitancy was observed

**Data availability statement:** The original data presented in the study are openly available through FigShare DOI: https://doi.org/10.6084/m9.figshare.31386241.

**Funding:** The author(s) received no specific funding for this work.

**Competing interests:** The authors have declared that no competing interests exist.

**Abbreviations:** WHO: World Health Organization; Dr.: Doctor; MCV: Measles Containing Vaccine; MOPH: Ministry Of Public Health; UNICEF: United Nations International Children's Emergency Fund; COVID-19: Coronavirus Disease 2019; CDC: Centers For Disease Control and Prevention; SDOH: Social Determinants of Health; SES: Socioeconomic Scale; DMS: Discrimination in Medical Setting Scale; VHL: Vaccine Health Literacy; DSES: Daily Spiritual Experience Scale; KMO: Kaiser–Meyer–Olkin; IRB: Institutional Review Board; LIU: Lebanese International University; SPSS: Statistical Package for Social Sciences; SD: Standard Deviations; ANOVA: Analysis of Variance; CI: Confidence Interval; VFC: Vaccines for Children; LMICs: Low- and Middle-Income Countries.

among individuals reporting higher discrimination in medical settings (Beta = 0.752, $p < 0.001$), stronger religious beliefs (Beta = 0.866, $p < 0.001$), and lower SES (Beta = 0.265, $p < 0.001$). Conversely, higher vaccine literacy (Beta = 0.216, p = 0.035) and a health education background (Beta = 6.262, p = 0.011) were associated with lower hesitancy.

## Conclusion

This study highlights key social determinants of vaccine hesitancy among a sample of Lebanese parents, emphasizing the role of medical discrimination, religious beliefs, and socioeconomic status. Improving vaccine literacy and health education, particularly in vulnerable populations, is crucial to reducing hesitancy. Addressing discrimination in healthcare settings and fostering trust in the medical system are also essential.

## 1. Introduction

Vaccines are widely regarded as one of the most successful public health interventions in history, saving millions of lives annually by preventing infectious diseases [1,2]. High vaccination coverage is essential for individual protection and herd immunity [3]. However, insufficient coverage can lead to the re-emergence of controlled diseases [4]. Of concern, Poliomyelitis, nearly eradicated in 125 countries, has resurged in conflict-affected regions such as Syria and Sudan [5,6].

Despite these successes, vaccine hesitancy, defined by the World Health Organization (WHO) as the delay in acceptance or refusal of vaccination despite the availability of vaccination services, is a significant global health challenge [2]. The WHO has identified vaccine hesitancy as one of the top ten threats to global health [7].

Misinformation spread through social media and anti-vaccination movements has exacerbated hesitancy, leading to declining vaccination rates globally [8,9]. In 2020, an estimated 23 million children worldwide did not receive essential childhood vaccines, marking the highest number since 2009 [10].

In Lebanon, low vaccination coverage has been linked to the resurgence of vaccine-preventable diseases, particularly measles and mumps [11,12]. In 2018, the coverage rates for the measles-containing vaccine (MCV) were 82% for the first dose (MCV1) and 63% for the second dose (MCV2), both below the WHO-recommended threshold of 95% [13]. Challenges include limited access to healthcare facilities, inadequate healthcare infrastructure, and insufficient awareness about vaccine schedules [11,12,14].

Social determinants of health (SDOH), defined as non-medical factors such as socioeconomic status, education, and cultural beliefs, play a significant role in shaping vaccination behaviors [15,16]. Factors contributing to hesitancy include cultural influences, gender, socioeconomic disparities, fear of adverse effects, skepticism about vaccine efficacy, and widespread misinformation [17–19]. In negative past

experiences and reluctance to vaccinate children have been identified as significant barriers, while effective communication and accurate knowledge have positively influenced parental decisions [20,21].

Despite these insights, there is a notable lack of research in Lebanon examining the social determinants of vaccine hesitancy among parents of children three years and younger. This study addresses this gap by identifying the social determinants of vaccine hesitancy among parents of children aged three years and below to inform targeted interventions to improve vaccination coverage and public health outcomes in Lebanon.

## 2. Methods

### Study design

An online observational cross-sectional study was conducted from August 22, 2024 to October 02, 2024, to comprehensively assess vaccine hesitancy among parents of children aged three years and younger residing in all Lebanese governorates.

This study focused on parents of children aged three years and below for several reasons: [1] This age group receives the majority of recommended childhood vaccines according to the Centers for Disease Control and Prevention (CDC) and Lebanese Ministry of Public Health vaccination schedules, including critical vaccines [8,22]; [2] Vaccine hesitancy decisions made during this period have the greatest impact on preventing vaccine-preventable diseases, as delayed or refused vaccinations during early childhood can lead to increased disease susceptibility and outbreaks [23]; [3] Previous research has shown that parental vaccine hesitancy is most pronounced during the first three years of life when vaccination schedules are most intensive [24]; and [4] Establishing positive vaccination behaviors during this critical period can influence parental attitudes toward future vaccinations throughout childhood and adolescence [23].

Parents were recruited using snowball sampling, a non-probability sampling technique appropriate for hard-to-reach populations [25]. Inclusion criteria were: [1] Lebanese parents or legal guardians, [2] having at least one child aged three years or below, [3] residing in Lebanon, and [4] ability to read and understand either Arabic or English.

The recruitment process began with an initial convenience sample of parents identified through personal networks. These initial participants were asked to share the survey link with other eligible parents in their networks via social media platforms (WhatsApp, Facebook, Instagram, LinkedIn). To maximize diversity, we specifically targeted parents from different governorates, socioeconomic backgrounds, and educational levels.

### Questionnaire

The questionnaire consisted of 39 closed-ended questions with pre-defined answers. It was divided into the following five sections:

The first section included the socio-demographic characteristics of participants, such as parents age, gender, marital status, area of residence, area of living (urban versus rural), university degree, type of education (health versus non-health), child age, gender, and birth order, presence of allergies, allergy to peanuts, in addition to the household crowding index (HCI), a measure used to assess residential overcrowding, calculated by dividing the number of people living in the house by the total rooms excluding kitchens and bathrooms [26].

The second section assessed vaccine hesitancy using a 23-item scale. Vaccine hesitancy was defined for participants as "a delay in acceptance or refusal of vaccines despite the availability of vaccine services." The scale measured two dimensions: vaccine hesitancy (16 items) and vaccine acceptance/trust (7 items). Respondents rated their level of agreement with statements concerning vaccination of children three years old and below using a 5-point Likert scale ranging from 1 (Strongly disagree) to 5 (Strongly agree). The vaccine hesitancy scale score was treated as a continuous variable (range: 23–115) in all analyses. Higher scores on the hesitancy subscale indicate greater vaccine hesitancy, while higher scores on the acceptance/trust subscale indicate more positive attitudes toward vaccination. The complete list of items and their factor loadings is presented in Table 1.

**Table 1. Factor analysis of vaccine hesitancy scale (Promax rotated component matrix).**

| Items | Factor 1 | Factor 2 |
|---|---|---|
| Concerned about components | 0.781 | – |
| Concerned about safety | 0.773 | – |
| Concerned about serious adverse effects | 0.768 | – |
| Fear of simultaneous multiple vaccines | 0.768 | – |
| Contracting a preventable disease is beneficial | 0.76 | – |
| Concerned about learning disabilities | 0.737 | – |
| Concerned about long lasting negative effect | 0.733 | – |
| Delay some vaccines | 0.728 | – |
| Delayed vaccination is better than not receiving | 0.723 | – |
| Benefit in contracting certain preventable diseases | 0.71 | – |
| Natural immunity is better than acquired | 0.704 | – |
| Diseases for which we vaccinate are not very prevalent | 0.698 | – |
| Healthy diets and lifestyles decrease the risk | 0.691 | – |
| Prefer to not put extra chemicals into child's body | 0.67 | – |
| Children do not need vaccines | 0.6 | – |
| New vaccines are risky | 0.571 | – |
| Vaccine in government program are beneficial | – | 0.882 |
| Vaccines are effective | – | 0.881 |
| Getting vaccines is protect my children | – | 0.88 |
| Child vaccination is important for others | – | 0.875 |
| Vaccines are important for child | – | 0.864 |
| Follow doctor recommendations | – | 0.858 |
| Information from the vaccine program is reliable and trustworthy | – | 0.803 |

Factor 1: vaccines hesitancy; Factor 2: vaccines trust. Kaiser-Meyer-Olkin (KMO) 0.919, Bartlett's test of sphericity <0.001, Percentage of variance explained 67.549%. Cronbach's alpha value: 0.920.

The third section encompassed the factors influencing childhood vaccination. It was divided into four sub-sections:

1. Source of information: participants were asked to indicate the first trusted source of information for encouraging child-hood vaccination. Among the options were healthcare professionals (i.e., the pediatrician, family doctor, pharmacist, and nurse), relatives, social and mass media advertisements, academic knowledge, the internet, and reading materials [27,28].

2. Level of influence: participants rated, on a scale of 1–10, the influence of the doctor, pharmacist and social media on childhood vaccination. Ratings from 1–3 indicated high influence; 4–7, moderate influence; and 8–10 low influence [27,28].

3. Religious concerns about childhood vaccination: participants were provided with four statements and were asked to rate them on a 5-point Likert scale ranging from strongly disagree to strongly agree [29,30]

   ● The animal-derived gelatin used in producing some vaccines as well as the human fetus tissue used in the rubella component

   ● I am worried of the possible porcine or non-halal ingredients content of vaccines

   ● I postpone my child vaccination during the religious fasting period

- I believe that the Lord will protect my child/children, no need for the vaccines

Attitude toward CDC-recommended vaccination: participants were asked to provide an answer regarding a list of recommended vaccines to a three-year old child and below [8]. The responses were grouped into three categories: (a) I refuse to give the vaccine to my child, (b) I prefer to delay vaccination, and (c) I am not sure about the vaccine and chose to trust doctor's opinion [7]. The listed vaccines were:

- Chickenpox vaccine (Varicella)
- Haemophilus influenzae b (Hib) Vaccine
- Hepatitis A vaccine
- Hepatitis B vaccine
- Influenza vaccine
- Polio vaccine
- Measles, Mumps, Rubella (MMR) vaccine
- Meningococcal vaccine
- Pneumococcal vaccine
- Rotavirus vaccine
- Tetanus, diphtheria, pertussis (DPT) vaccine

The fourth section evaluated the SDOH of vaccines hesitancy. It consisted of four scales:

1. The Socioeconomic Scale (SES) designed to capture respondent financial strains consists of 8 items. Participants were asked to rate their financial stress over the past month on a scale from 1 to 10, covering areas such as financial stress, satisfaction with financial situation, ability to meet living expenses, and financial stability. Higher scores indicated lower financial stress and better financial well-being. The total possible score ranged from 8 to 80. The scale demonstrated strong internal consistency (Cronbach's alpha = 0.929) [31].

2. The Discrimination in Medical Setting Scale (DMS) was used to evaluate patients' self-reported experiences of discrimination in healthcare environments. The scale includes 7 items rated on a 5-point Likert scale (1 = Never to 5 = Always), with higher scores reflecting greater perceived discrimination. The total possible score range is 7–35. The scale demonstrated strong internal consistency (Cronbach's alpha = 0.879). Participants reported the frequency of discrimination experiences related to their clinical conditions during medical interactions [32].

3. The Vaccine Health Literacy (VHL) Scale was used to assess parents' ability to obtain, understand, and use vaccine-related information to make informed vaccination decisions for their children. The scale consisted of 14 items divided into three subscales: functional literacy (understanding basic vaccine information), communicative literacy (ability to extract and apply vaccine information), and critical literacy (ability to critically analyze vaccine information). Responses were assessed using a 4-point Likert scale (1 = Never, 2 = Rarely, 3 = Sometimes, 4 = Often), with total scores ranging from 14 to 56. Higher scores indicated better vaccine health literacy. The scale demonstrated good internal consistency (Cronbach's alpha = 0.819) [33,34].

4. The Daily Spiritual Experience Scale (DSES) was used to assess parents' spiritual and religious experiences in daily life. The scale consists of 6 items measuring the frequency of spiritual experiences such as feeling God's presence, finding strength in religion, experiencing inner peace, desiring closeness to God, feeling God's love, and being

spiritually touched by creation. Responses were rated on a 6-point Likert scale (1 = Many times a day, 2 = Every day, 3 = Most days, 4 = Some days, 5 = Once in a while, 6 = Never or almost never), with lower scores indicating more frequent spiritual experiences. The total possible score ranged from 6 to 36. The scale demonstrated excellent internal consistency (Cronbach's alpha = 0.928) [35].

## Data collection

Data were collected using an online questionnaire formulated in English and created on Google Forms, a cloud-based survey powered by Google™. A forward translation was first performed from English to Arabic, and then back-translated to English. The two English versions were compared, with minor discrepancies corrected by consensus between the translators and the authors. A pilot-testing of the questionnaire included 20 parents from different gender, socio-economic and educational backgrounds to identify any potential ambiguities in language or expressions. Subsequent feedback prompted necessary revisions for clarity and cultural relevance. The questionnaire was electronically distributed using the snowball sampling technique [25]. The link to the questionnaire was shared using various social media platforms, including WhatsApp, Facebook, Instagram, and LinkedIn.

## Ethical approval

This study, approved by the Institutional Review Board at the Lebanese International University under code 2024ERC-020-LIUSOP, adhered to the Declaration of Helsinki [36]. Before filling out the online survey, participants were briefed about the study objectives and their right to withdraw at any time.

Informed consent was obtained from all participants through an explicit consent process at the beginning of the online survey. Before accessing the questionnaire, participants were presented with a detailed informed consent statement explaining: [1] the study's purpose and objectives, [2] voluntary nature of participation, [3] right to withdraw at any time without consequences, [4] data confidentiality and anonymization procedures, [5] absence of financial compensation, and [6] contact information for questions. Participants provided informed written consent by reading the consent statement and actively clicking a checkbox labeled 'I have read and understood the above information and agree to participate in this study.' This action was required to proceed to the questionnaire, ensuring documented agreement. The checkbox action was recorded electronically and time-stamped. Participants who did not check the consent box could not access the survey.

This study surveyed parents about their vaccine-related attitudes and behaviors; no data were collected directly from children, therefore parental consent for child participation was not applicable. Parents provided consent for their own participation in discussing their children's vaccination." Collected data were encrypted, stored in password-protected computers, and presented as de-identified electronic files in Microsoft Excel and Statistical Package for Social Sciences (SPSS), version 26.

## Sample size calculation

The minimal sample size was calculated using Epi-Info version 7.2.6. Based on a previously published study [24], the prevalence of vaccine hesitancy among parents with children less than three years was 20.2%. The minimum necessary sample was n = 238 participants considering an alpha error of 5%, and a power of 95%.

## Statistical analysis

Data were analyzed using SPSS. A descriptive analysis was performed using absolute frequencies and percentages for categorical variables and means and standard deviations (SD) for quantitative measures. Bivariate analysis was conducted to examine the association between vaccine hesitancy (dependent variable) and SDOH and other independent

variables. For categorical variables, Chi-square tests were used, and for continuous variables, the student's t-test was applied. A p-value < 0.05 was considered statistically significant.

To examine the vaccine uptake patterns across different clusters of respondents, an ANOVA (Analysis of Variance) was conducted. The data was grouped into three clusters based on participants' attitudes toward vaccination: Cluster 1 (refuse to give the vaccine to their child), Cluster 2 (prefer to delay vaccination), and Cluster 3 (trust the doctor's opinion without having prior knowledge of the vaccine). The ANOVA test was used to assess the differences in vaccine uptake across these three clusters, and post hoc analyses (Tukey HSD) were performed to identify specific pairwise differences. The significance level was set at p < 0.001.

A linear regression was employed with vaccine hesitancy scale as the dependent variable. Independent variables with a p-value < 0.05 from the bivariate analysis were included in the model using the Enter method. The Omnibus test was performed to assess model significance, while the correlation matrix was examined to check for collinearity. The Hosmer and Lemeshow test were used to evaluate the model's goodness-of-fit, and Nagelkerke $R^2$ was reported to determine the explanatory power of the model. Scale reliability was assessed using Cronbach's alpha. The significance level was set at P < 0.05 with a confidence interval (CI) of 95%.

## 3. Results

### Reliability and validity of the vaccine hesitancy scale

A factor analysis was conducted on responses from 251 participants to explore childhood vaccine hesitancy. The Kaiser–Meyer–Olkin measure indicated excellent sampling adequacy (KMO = 0.919), and Bartlett's test supported the factorability of the correlation matrix ($\chi^2$ (253) = 6339.755, p < .001). Factor extraction was based on eigenvalues greater than 1 (Kaiser criterion). The analysis yielded two major components explaining 67.549% of the total variance. The first component (vaccine hesitancy) accounted for 37.948% of the variance (eigenvalue = 8.728), while the second component (vaccine acceptance/trust) explained an additional 29.600% (eigenvalue = 6.808).

Factor loadings ranged from 0.571 to 0.781 for Factor 1 and 0.803 to 0.882 for Factor 2, with no significant cross-loadings (all cross-loadings < 0.40). The correlation between the two components was relatively weak (r = 0.105), supporting the discriminant validity of the two factors. Reliability analysis yielded a Cronbach's alpha of 0.920, indicating excellent internal consistency. The scale demonstrated appropriate validity and reliability (Table 1).

Table 2 outlined the socio-demographic characteristics of participants. The study included 251 participants with a female predominance (55.8%) and a mean age of parents of 34.94 ± 8.69 years, while the mean child's age was 21.35 ± 11.53 months. The majority of participants (58.6%) were residing in Bekaa region, predominantly living in urban areas (72.1%). Educational characteristics revealed that 67.3% did not have a university degree. Among all participants, only 13.1% had health-related education. Regarding their children, 45% were first-born, while 55% were second-born or later, predominantly male (55.8%) children with limited number of children (13.5%) suffering from allergies. Peanut allergies affected 5.6% of the total sample.

Fig 1 shows that pediatricians are by far the most significant source of information at 43.7%, followed by both social media and mass media advertisements tied at 19.8%. Internet sources come in at 10.7%, while the remaining sources each represent about 1% or less of the total.

Fig 2 reveals important insights about religious concerns regarding childhood vaccination. The results showed varying levels of agreement across different religious considerations. Most respondents expressed "neutrality or disagreement" positions regarding "animal-derived gelatin and human fetus tissue" in vaccines. Similar patterns were observed for concerns about "porcine or non-halal ingredients". "Postponing vaccinations during religious fasting periods" exhibited notable spread across all response categories, though "disagreement" remained predominant. The "belief that divine protection alone is sufficient for a child's health" also showed mixed responses, with stronger representation in both "agreement and disagreement" categories.

**Table 2. Socio-demographic characteristics of participants.**

| Variable | N (%) |
|---|---|
| **Total participants** | **251 (100)** |
| **Gender** | |
| Males | 111 (44.2%) |
| Females | 140 (55.8%) |
| **Area of residence** | |
| Beirut | 48 (19.1%) |
| Mount Lebanon | 18 (7.2%) |
| Bekaa | 147 (58.6%) |
| Other regions | 38 (15.1%) |
| **Area of living** | |
| Urban | 70 (27.9%) |
| Rural | 181 (72.1%) |
| **University degree** | |
| No | 169 (67.3%) |
| Yes | 82 (32.7%) |
| **Type of education** | |
| Health education | 33 (13.1%) |
| Non-health education | 218 (86.9%) |
| **Childbirth order** | |
| First | 113 (45%) |
| Second and above | 138 (55%) |
| **Child gender** | |
| Male | 140 (55.8%) |
| Female | 111 (44.2) |
| **Child allergy** | |
| No | 217 (86.5%) |
| Yes | 34 (13.5%) |
| **Peanut allergy** | |
| Yes | 14 (5.6%) |
| No | 237 (94.4%) |
| **Variable** | ***Mean±SD*** |
| **Parent's Age** | 34.94±8.69 |
| **Child's Age** | 21.35±11.53 |
| **HCI** | 1.08±0.50 |

SD: standard deviation; Household crowding index: HCI.

Table 3 presents the cluster analysis, which identified three distinct groups among the 251 participants: Cluster 1 (I refuse to give this vaccine to my child), comprising 115 participants (45.8%); Cluster 2 (I prefer to delay giving this vaccine to my child), with 3 participants (1.2%); and Cluster 3 (I have no idea about this vaccine; I just trust the doctor's opinion), the largest group with 133 participants (53.0%).

The ANOVA results revealed statistically significant differences in vaccine uptake across the three clusters for all vaccines tested (p<0.001). The F-statistics were large for all vaccines, suggesting that the differences between clusters were much greater than the within-cluster variance. The most pronounced differences were observed for the Polio

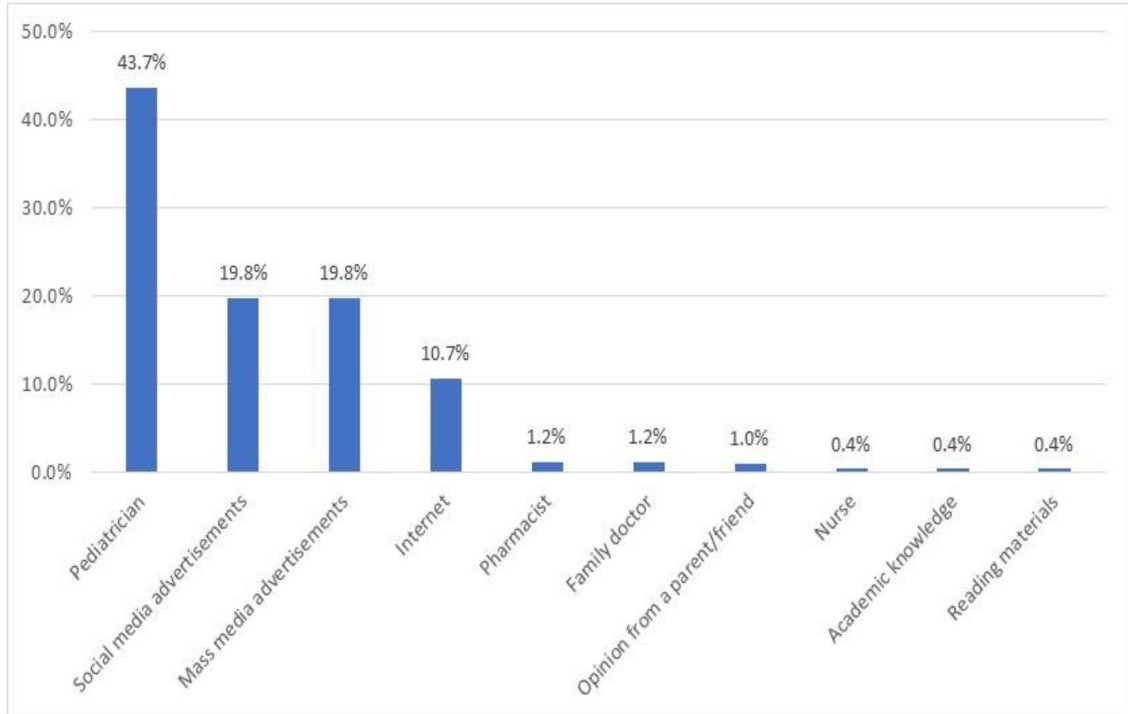

**Fig 1. First trusted source of information.**

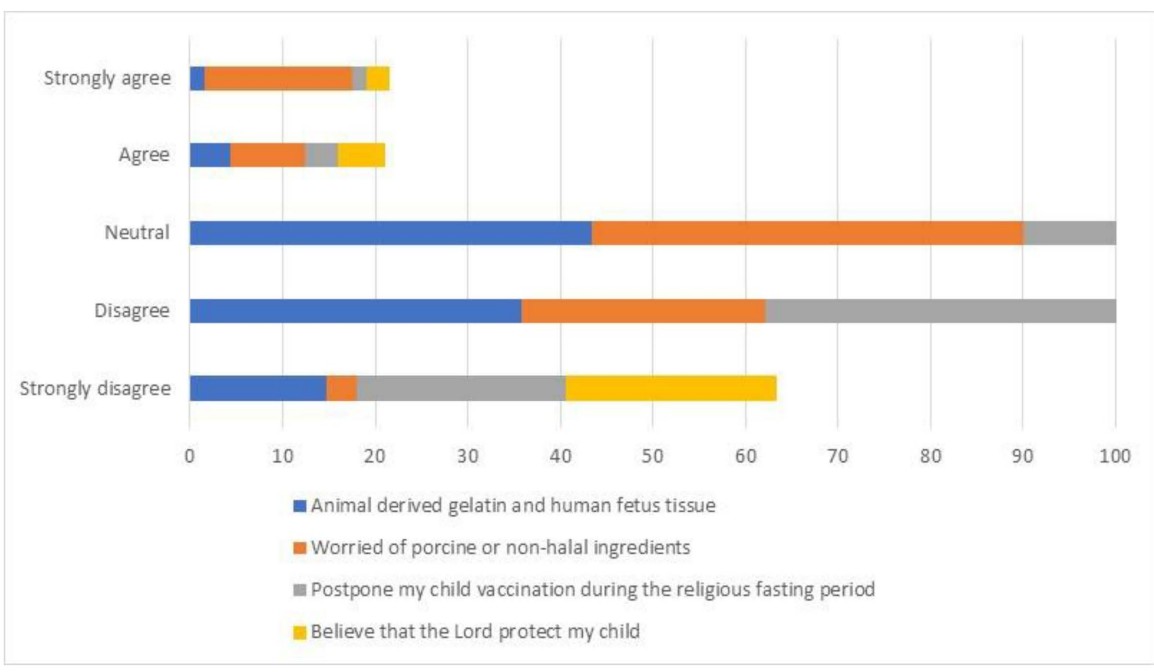

**Fig 2. Religious concerns concerning childhood vaccination.**

**Table 3. ANOVA results for vaccine uptake across clusters.**

| Vaccine type | F-Statistic | p-Value |
|---|---|---|
| chickenpox | 2019.497 | <0.001* |
| hib | 112.536 | <0.001* |
| hepatitis a | 115.277 | <0.001* |
| hepatitis b | 1600.906 | <0.001* |
| influenza | 337.566 | <0.001* |
| polio | 4238.523 | <0.001* |
| mmr | 2058.556 | <0.001* |
| meningococcal | 2069.408 | <0.001* |
| pneumococcal | 2781.886 | <0.001* |
| rotavirus | 1668.465 | <0.001* |
| DPT | 2728.805 | <0.001* |

*$p < 0.05$ is significant.

vaccine (F = 4238.523), followed by Pneumococcal (F = 2781.886) and DPT vaccines (F = 2728.805). While still significant, the differences for the Hepatitis A (F = 115.277) and Hib vaccines (F = 112.536) showed relatively lower variation between clusters.

Cluster 1 (those who refuse vaccines) exhibited the lowest uptake across all vaccines, while Cluster 3 (those who trust the doctor's opinion) showed the highest vaccine uptake. Cluster 2 (those who prefer to delay vaccination) displayed moderate uptake levels.

Table 4 revealed significant differences in vaccine hesitancy scores across different socio -demographic variables. Participants with health education demonstrated significantly lower vaccine hesitancy scores (55.00 ± 13.96) than those with non-health education (63.52 ± 13.66) with p = 0.001. While not reaching statistical significance, several other demographic factors showed variations in hesitancy scores: participants with university degrees showed lower hesitancy scores (60.68 ± 14.05) than without (63.24 ± 13.90), and urban residents demonstrated lower scores (60.70 ± 14.28) than rural residents (63.06 ± 13.84). Parents of children with peanut allergies showed lower hesitancy scores (56.07 ± 16.58) compared to those without (62.78 ± 13.75) with marginally significant difference. Other factors including gender, childbirth order, child's gender, and general child allergies showed minimal differences in vaccine hesitancy scores.

Fig 3 illustrates the varying levels of influence that doctors, pharmacists, and social media have on vaccine hesitancy decisions. Moderate influence was the most common response across all three sources, consistently reaching about 50% or higher. Healthcare professionals (doctors and pharmacists) had very low percentages in the "low influence" category (3.6% and 5.2% respectively), while social media had a slightly higher proportion of low influence responses (9.2%). Doctors maintained the highest percentage of "high influence" ratings (46.2%) compared to pharmacists (37.5%) and social media (39.8%), suggesting they remain the most influential source for vaccination decisions.

Table 5 outlines the correlation analysis examining factors associated with vaccine hesitancy. Results indicate that individuals who experienced more discrimination were more likely to express vaccine hesitancy (r = 0.287, p < 0.001). Religious beliefs showed the second strongest correlation (r = 0.180, p < 0.01). In addition, vaccine health literacy (r = 0.175, p < 0.01) and socioeconomic status (r = 0.155, p < 0.05) also exhibited significant positive correlations with vaccine hesitancy. Interestingly, the HCI, parent's age, and child's age showed no significant correlations with vaccine hesitancy.

Table 6 reveals several key factors significantly influencing vaccine hesitancy. Higher level of literacy regarding vaccines is correlated with an increased likelihood of accepting vaccination (Beta = 0.216, p = 0.035). Similarly, results indicated that individuals who perceive higher levels of discrimination are more likely to exhibit vaccine hesitancy (Beta = 0.752, p < 0.001).

**Table 4. Factors affecting childhood vaccine hesitancy as dependent variable.**

| Variables | Vaccine hesitancy scale | P-value |
|---|---|---|
| | *Mean±SD* | |
| **Gender** | | |
| Male | 62.02 ± 14.68 | 0.699 |
| Female | 62.71 ± 13.42 | |
| **Area of living** | | |
| Urban | 60.70 ± 14.28 | 0.231 |
| Rural | 63.06 ± 13.84 | |
| **University degree** | | |
| Yes | 60.68 ± 14.05 | 0.175 |
| No | 63.24 ± 13.90 | |
| **Type of education** | | |
| Health education | 55.00 ± 13.96 | 0.001* |
| Non-health education | 63.52 ± 13.66 | |
| **Childbirth order** | | |
| First | 62.78 ± 14.06 | 0.7 |
| Second and above | 62.09 ± 13.94 | |
| **Child's Gender** | | |
| Male | 61.12 ± 16.07 | 0.103 |
| Female | 64.01 ± 10.61 | |
| **Child allergies** | | |
| Yes | 62.73 ± 14.90 | 0.882 |
| No | 62.35 ± 13.86 | |
| **Peanut allergy** | | |
| Yes | 56.07 ± 16.58 | 0.081 |
| No | 62.78 ± 13.75 | |

Independent samples t-test for continuous variables; Chi-square test for categorical variables; *p < 0.05 is significant.

Religious belief demonstrated a significant positive association (Beta = 0.866, p < 0.001), implying that individuals with stronger religious beliefs tend to exhibit greater vaccine hesitancy. SES further emerged as a critical determinant, with a significant positive effect (Beta = 0.265, p < 0.001), suggesting that individuals from lower socioeconomic backgrounds are more prone to vaccine hesitancy. Finally, the type of education (health vs. non-health) shows a significant difference (Beta = 6.262, p = 0.011), with individuals educated in health-related fields demonstrating lower levels of vaccine hesitancy.

## 4. Discussion

The findings of this study highlight the complex interplay of social determinants influencing childhood vaccine hesitancy among Lebanese parents. Socioeconomic status (SES), discrimination in medical settings, religious beliefs, and vaccine health literacy (VHL) were significant factors shaping parental attitudes toward vaccination. These findings align with global trends recognizing vaccine hesitancy as a multifaceted issue influenced by cultural, social, and economic contexts.

**Key determinants of vaccine hesitancy**

**Socioeconomic status and discrimination.** Lower SES was strongly associated with higher vaccine hesitancy, consistent with studies from other low- and middle-income countries (LMICs) where financial constraints and limited

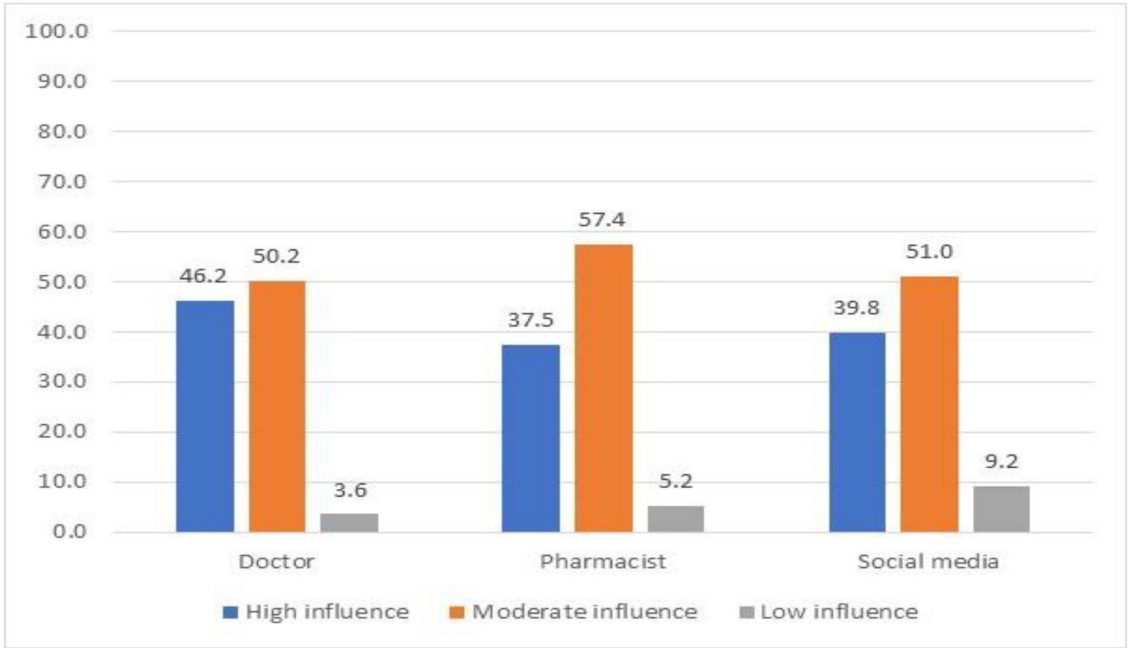

**Fig 3. Factors influencing childhood vaccination.**

**Table 5. Correlation between childhood vaccine hesitancy and SDOH.**

| Variable | Correlation Coefficient (r) | Significance Level |
|---|---|---|
| DMS | 0.287 | P<0.001* |
| DSES | 0.18 | 0.004* |
| VHL | 0.175 | 0.005* |
| SES Scale | 0.155 | 0.014* |
| HCI | −0.05 | 0.432 |
| Age | 0.068 | 0.285 |
| Child age | 0.104 | 0.1 |

*p<0.05: Significant value

SES: socio-economic scale; HCI: household crowding index; DIS: discrimination scale; VHL: vaccine health literacy.

HCI: Household Crowding Index (number of people/number of rooms excluding kitchen and bathroom.

access to healthcare services exacerbate hesitancy [37]. Discrimination in medical settings further compounded this issue, with parents reporting higher discrimination more likely to distrust healthcare systems and delay or refuse vaccinations. Research examining health disparities demonstrated that even in settings where healthcare is provided free at the point of use, structural and institutional barriers within health systems significantly contribute to vaccine hesitancy [38]. Studies have documented systematically lower access to primary, inpatient, and outpatient care among populations experiencing healthcare inequities, coupled with higher likelihood of experiencing adverse encounters with healthcare systems, suggesting that healthcare access barriers and discriminatory practices rather than financial constraints alone drive hesitancy disparities [38]. This finding indicates the need for healthcare providers to address

**Table 6. Linear regression considering Vaccine hesitancy as dependent variable.**

| Variable | Unstandardized Beta | Standardized Beta | p-value | 95% CI | |
|---|---|---|---|---|---|
| | | | | Lower Bound | Upper Bound |
| VHL | 0.216 | .123 | 0.035* | .015 | .418 |
| DMS | 0.752 | .325 | <0.001* | .469 | 1.035 |
| DSES | 0.866 | .201 | <0.001* | .386 | 1.347 |
| SES | 0.265 | .262 | <0.001* | .143 | .388 |
| Type of education (Health*/Non-health) | 6.262 | .152 | 0.011* | 1.425 | 11.100 |

*Significant value.

SES: socio-economic scale; DMS: discrimination scale; VHL: vaccine health literacy; DSES: Daily Spiritual Experience Scale.

systemic biases through culturally sensitive communication and equitable care [40]. Healthcare authorities must actively involve marginalized communities in the development and implementation of healthcare initiatives to build trust and foster shared ownership, with participatory approaches proving effective in reducing perceptions of imposed interventions and promoting community buy-in [38].

**Religious beliefs.** Religious concerns, particularly regarding vaccine ingredients (e.g., animal-derived gelatin or non-halal components), played a significant role [39,40]. While most respondents expressed neutrality or disagreement with these concerns, a notable minority cited religious beliefs as a barrier [41]. Tailored interventions, such as engaging religious leaders in vaccine advocacy and providing transparent information about vaccine ingredients, could help address these concerns, as demonstrated in other Muslim-majority countries [41].

Consistent with systematic reviews and meta-analysis, socioeconomic disparities and health literacy are universal determinants of vaccine hesitancy [42,43]. A global meta-analysis found parental vaccine hesitancy prevalence of 21.1% across over 30 countries, with lower-middle-income countries exhibiting rates ranging from 3.4% to 41.6%, reflecting substantial variability across geographic and socioeconomic contexts [42,44]. Unfavorable attitudes toward vaccination, including reduced perceptions of vaccine effectiveness and mistrust of health authorities, constitute major barriers to vaccine uptake across diverse settings [44]. However, the prominence of discrimination and religious beliefs (Beta = 0.866, p < 0.001) appears more pronounced in Lebanon, necessitating locally adapted public health strategies rather than Western imported approaches.

**Vaccine health literacy.** Higher VHL was associated with lower vaccine hesitancy, emphasizing the importance of health education in combating misinformation. Evidence from systematic reviews and multi-country studies confirms that individuals with high levels of vaccine literacy are significantly less likely to be unvaccinated, with this association remaining robust even after controlling for socioeconomic confounding factors [45,46]. The pathway through which vaccine literacy impacts hesitancy operates partially through mediating factors including confidence in vaccines, conspiracy beliefs, and complacency [46]. Parents with a health education background were significantly less hesitant, suggesting that targeted educational campaigns could improve vaccine acceptance [33]. Systematic reviews provide strong evidence that higher health literacy is inversely associated with vaccine hesitancy across diverse contexts, supporting public health approaches that prioritize increasing health literacy as a promising strategy for tackling vaccine hesitancy and reducing the burden of vaccine-preventable diseases [45].

**Healthcare providers influence.** Pediatricians were the most trusted source of vaccine information (43.7%), similar to findings in Syria [47]. Cluster analysis reinforced that healthcare provider recommendations significantly impact vaccine decisions. The moderate influence of pharmacists suggests an untapped potential for expanding their role in vaccination programs. A systematic review demonstrated that pharmacist involvement consistently increased vaccine coverage [48]. In Lebanon, legal reforms enabling pharmacists to administer vaccines could improve access, particularly in underserved

                                                                     

areas. Our findings also aligned with a Turkish study where higher education was associated with lower hesitancy, suggesting important contextual differences across Middle Eastern populations [49].

Parents with health-related education background displayed significantly lower hesitancy, aligning with previous research showing that medical knowledge increases vaccines confidence [50]. Parents generally held positive attitudes toward all mandatory and recommended vaccines in Lebanon, consistent with a 2020−2021 national study [51]. These insights reinforce the need to study did not directly assess COVID-19-specific attitudes or experiences, several findings suggest that pandemic-related trust erosion may have influenced our results: [1] the heightened discrimination scores may reflect negative COVID-19 vaccine rollout experiences, [2] a 19.8% reliance on social media as a primary information source is concerning given social media's role in amplifying misinformation, and [3] the association between lower SES and vaccine hesitancy may have been amplified by the pandemic's disproportionate economic impact on vulnerable populations [52,53].

**The role of social media in vaccine decision-making.** Social media platforms significant sources of vaccine misinformation, with research demonstrating that non-scientific, sensational, or manipulative content on these platforms negatively affects public attitudes toward vaccines [49]. Social media misinformation are identified as a key factor influencing vaccine hesitancy in their systematic review of parents in Saudi Arabia, with vaccine hesitancy rates varying from 7.1% to 72.2% [43]. The mechanisms through which social media influences vaccine decisions are multifaceted: these platforms lack editorial oversight and frequently promote unverified, emotionally-charged claims about vaccine safety and side effects that have no scientific basis. Cagnotta (2025) explain that social media rapidly exposes users to both factual and emotional content, thus facilitating the spread of fear, doubt, and mistrust [54]. Misinformation regarding side effects, particularly when disseminated through digital and social media, can significantly distort individuals' perceptions [49]. Algorithmic amplification creates echo chambers where vaccine-skeptical beliefs are continuously reinforced, making users less likely to encounter evidence-based information. Misinformation and fake news shared on social media could influence parents' attitudes toward children's vaccination, with findings showing that female, more affluent, and more engaged social media users exhibited greater levels of vaccine hesitancy [55]. Research has demonstrated that misinformation spreads significantly faster than corrections, and visually compelling but misleading content, including emotional testimonials, anecdotes about alleged vaccine injuries, and conspiracy theories, tends to be more engaging and shareable than factual, science-based information. However, it is important to acknowledge that social media can also serve as a positive platform for disseminating accurate health information when properly leveraged. Evidence-based educational approaches are necessary to combat vaccine misinformation effectively. Cagnotta (2025) recommend a multi-pronged communication approach to address parental vaccine hesitancy: first, disseminating accessible health information using clear language and transparent data sources; second, deploying educational initiatives across schools, medical facilities, and digital platforms that encourage participation from communities and healthcare workers; third, organizing dialogue sessions that validate parental concerns while providing accurate information; and fourth, systematically identifying and refuting false claims through ongoing surveillance and fact-based responses. Studies using rigorous trial methodologies have confirmed that educational programs significantly reduce vaccine reluctance among parents [49,54]. The manner in which vaccine information reaches parents proves critical for closing knowledge deficits and countering the swift dissemination of false narratives prevalent on social platforms. Communication strategies to enhance targeted communication include education through healthcare providers and dispelling social media misinformation, which are considered essential. Without concerted counter-misinformation efforts, improvements in other determinants may have limited impact. Given Lebanon's high social media penetration (19.8% of parents in our study identified social media as their primary trusted source) and demonstrated influence on vaccine decisions, addressing social media-driven hesitancy should be a top priority. Educational strategies must be well-designed and evidence-based to effectively counter vaccine misinformation. Communication strategies to enhance targeted communication include education through healthcare providers and dispelling social media misinformation, which are

considered essential [54]. Without concerted counter-misinformation efforts, improvements in other determinants may have limited impact.

## Intervention strategies

Our findings support multifaceted interventions: [1] Legislation mandating vaccinations for school entry, as demonstrated in California [56,57]; [2] Educational initiatives for healthcare providers and the public, shown to enhance confidence among hesitant parents [58]; [3] Mobile health clinics reaching underserved populations [59]; and [4] Free or subsidized vaccinations, exemplified by the U.S. Vaccines for Children program [60]. Implementing these strategies coordinately can address vaccine hesitancy's complex nature and improve public health outcomes.

## Limitations

This study has several limitations that should be considered. As a cross-sectional study, it is subject to selection bias, limiting causal inferences. The use of a self-reported questionnaire introduces the provided a more comprehensive understanding of vaccine hesitancy and improved of a self-reported questionnaire introduces the risk of social desirability bias, as participants may have provided responses they perceived as more socially acceptable rather than their true beliefs.

Additionally, while snowball sampling allowed us to reach a geographically diverse sample across all Lebanese governorates, this method may introduce selection bias as participants with stronger social networks and internet access were more likely to be recruited. We acknowledge that this may have excluded parents from the most disadvantaged populations with limited digital literacy or internet connectivity. The refusal rate could contribute to selection bias, as parents who declined participation could not be compared with those who agreed to participate.

Moreover, the study was conducted online, excluding individuals without internet access or those less proficient in technology, which may have affected the representativeness of the sample. Although multiple confounding variables were accounted for, some may have been overlooked, leaving the findings susceptible to residual confounding bias. Another limitation is that the study focused only on Lebanese citizens, excluding non-Lebanese residents living in Lebanon. Including this population could have provided a more comprehensive understanding of vaccine hesitancy and improved the generalizability of the findings. Furthermore, data collection took place during a period of economic and security instability in Lebanon, which may have influenced participants' perceptions and their access to healthcare services.

Despite these limitations, the study has several notable strengths. The validation of the scales before data collection ensured the reliability of the measurements. Additionally, this is the first study of its kind conducted among the Lebanese population, specifically focusing on the selected age group, providing valuable insights into vaccine hesitancy within this context. The inclusion of participants from various regions across Lebanon further strengthened the validity of the results. Importantly, the use of an online questionnaire facilitated access to individuals in rural areas, improving the study's reach.

## Future research directions

Given these findings, further research is needed to address the limitations identified, particularly by incorporating a larger and more diverse sample, including non-Lebanese residents, and employing longitudinal study designs to better assess causal relationships. Future studies should also explore the impact of external factors, such as economic conditions and healthcare accessibility, on vaccine hesitancy to provide a more comprehensive understanding of this issue. Additionally, research should focus on the role of pharmacists in vaccination programs, not only in childhood immunization but also in administering booster vaccines for adults.

## 5. Conclusion

This study highlights the multifaceted nature of childhood vaccine hesitancy among Lebanese parents, shaped by socio-economic status, discrimination in medical settings, religious beliefs, and vaccine health literacy. Healthcare professionals

play a crucial role in shaping vaccine attitudes. Pediatricians were the most trusted source of vaccine information, while pharmacists, despite their potential, remain underutilized. Globally, pharmacist-led vaccination initiatives have expanded immunization coverage, suggesting that integrating them into vaccination programs could improve accessibility. The concerning reliance of nearly one in five parents on social media as their primary vaccine information source, combined with the potential lingering effects of COVID-19-related distrust, underscores the urgent need for targeted interventions that address both digital misinformation and institutional trust deficits. A multi-faceted approach combining public health strategies, education, social media counter-messaging, and stronger healthcare provider involvement is essential to overcoming vaccine hesitancy and improving immunization rates in Lebanon.

## Institutional review board statement

The study was conducted in accordance with the Declaration of Helsinki, and approved by the Institutional Review Board at the Lebanese International University under code 2024ERC-020-LIUSOP.

## Informed consent statement

Informed consent was obtained from all subjects involved in the study.

## Author contributions

**Conceptualization:** Reem Awad, Katia Iskandar.

**Data curation:** Reem Awad, Anna-Maria Henieneh, Robin Farah, Michelle Cherfane, Reham Kotb, Katia Iskandar.

**Formal analysis:** Reem Awad, Pascale Salameh, Michelle Cherfane, Katia Iskandar.

**Methodology:** Reem Awad, Robin Farah, Katia Iskandar.

**Supervision:** Reem Awad, Katia Iskandar.

**Validation:** Reem Awad, Katia Iskandar.

**Visualization:** Reem Awad, Katia Iskandar.

**Writing – original draft:** Reem Awad, Katia Iskandar.

**Writing – review & editing:** Reem Awad, Mohamad Rahal, Anna-Maria Henieneh, Pascale Salameh, Robin Farah, Michelle Cherfane, Reham Kotb, Diana Nakhoul, Rêve Khadaj, Nayla Habib, Katia Iskandar.

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
