## [Decision Letter · Decision Letter 0]

12 Sep 2025

Dear Dr. Awad,

The following changes are required before submission:

In the Methods section, please describe the vaccine hesitancy scale and include linear regression as the statistical procedure used to create Table 6.

Cite the references for each scale and provide a brief description.

Expand the description of the factor-analytic approach, specify the model, and cite the references.

Describe all derived variables and thresholds, explain how you chose the cut-points, and clarify the directionality.

In the results section, insert the tables near the corresponding texts.

Replace "their" by "my" in the sentence "I refuse to give the vaccine to their child", in the Methods.

State the purpose of VHL scale as done for the other ones.

The following changes are recommended:

Justify the reasons age group three or less was selected.

Describe how the parents were selected.

Compare the results with similar or other countries.

Discuss the reliance of parents on social media for their medical information.

Report reliability statistics (e.g., Cronbach’s alpha) in the Results rather than in the Methods.

Present Methods and Results in a stepwise manner aligned with measurement variables.

This academic editor strongly requires that you shorten the manuscript, especially the Introduction and Discussion.

Change the order of Tables 1 and 2.

Correct the following typos: In the Questionnaire, by the end of the paragraph, write parents' religious beliefs instead of believes; in the fourth sentence of Data Collection, modify as "A pilot-testing of the questionnaire included 20 parents from different gender, socio-economic and educational backgrounds"

Results:

Provide the response rate.

Modify the first line of second paragraph as ""Figure 1 shows that pediatricians are by far the most significant source of information"

Five lines below, change neutral or disagreement to neutrality or disagreement.

ANOVA results: write the first letter of polio, pneumococcal, and hepatitis in lower case.

Table 5 description: define HCI.

We look forward to receiving your revised manuscript.

Kind regards,

Vladimir Berthaud

Academic Editor

PLOS ONE

Journal Requirements:

5. Please include your tables as part of your main manuscript and remove the individual files. Please note that supplementary tables (should remain/ be uploaded) as separate "supporting information" files

Additional Editor Comments:

Reviewer #1: Minor revision

Review Comments to the Author

Awad et al explore the social determinants influencing childhood vaccine hesitancy among a sample of Lebanese parents, focusing on how individuals are born, grow, live, and work.

1) Why did the authors choose the age group <=3 years of age? Why not broaden the age group?

2) How were the parents selected?

3) Discussion-how do these results compare to other countries and their beliefs? This is important to generalize the results to other countries.

4) Discussion-was there a factor of the distrust that developed during COVID influence some of these beliefs?

5) The results reveal that many parents are dependent on social media for their medical information-please expand on this as this is concerning and how can this be changed?

Reviewer #2: Minor revision

ABSTRACT

1) There was mentioned about involving questionnaire on 'political and economic instability’ but not mentioned in the Methods or any other sections of the main text.

METHODS

1) in the sentence "I refuse to give the vaccine to their child", should the word "their" actually be replaced with "my"?

2) Give the full term for SDOH as it is first mentioned in the main text, besides in the abstract.

3) It was mentioned that the SES scale was used to capture respondent financial strains, DMS was to evaluate patients' self-reported experiences of discrimination in healthcare environments, DSES was used to examine parents' religious beliefs. But the purpose of VHL scale was not stated. Please provide at least a sentence that clearly mention the use of VHL scale.

4) It was stated that vaccine hesitancy scale was used as the dependent variable. But such scale was not described in the method. Please provide methods in determining the dependent variable. If the vaccine hesitancy scale was employed as claimed in the paper, clearly describe the scale like how other scales were described. Also, include explanation on the outcomes of vaccine hesitancy. Are the outcomes in binary outcomes since logistic regression was conducted? Or the outcomes are continuous, like the higher the score, the higher the hesitancy? If they are continuous, shouldn't Linear regression be performed rather than logistic regression?

RESULTS

1) What statistical test was used for results in Table 4? Perhaps can state in the table's footnote.

2) Linear regression was used in Table 6, but the method was not mentioned in the Methods section, instead Logistic regression was mentioned.

3) Why other factors like sociodemographic factors were not included in the analysis of Linear regression in Table 6? Is there any specific reason?

Reviewer #3: Reject

Question 4: Is the manuscript presented in an intelligible fashion and written in standard English? No

Review Comments to the Author

Thank you for the opportunity to review this manuscript.

I appreciate the authors’ contribution to the vaccine hesitancy literature. However, several issues should be addressed to strengthen the paper.

Methods

- Questionnaire provenance and content: Please clarify whether the questionnaire was newly developed or adapted from existing instruments. Validation Provide citations and brief descriptions for each scale (e.g., SES, DMS, VHL), including example items or domains, scoring, and interpretation. Define all abbreviations at first use.

- Factor-analytic approach: Clearly describe the factor methods used, including whether the analysis was exploratory or confirmatory. Specify the factor model, provide (or reference) a path/construct diagram, and report key details (extraction and rotation methods, factor retention criteria, factor loadings, cross-loadings, and—if CFA—model fit indices such as CFI/TLI/RMSEA/SRMR).

- Variable definitions and cut-points: Define all derived variables and thresholds (e.g., “high/low” discrimination scale, vaccine health literacy). Explain how cut-points were chosen (prior literature vs data-driven) and ensure directionality is clear so readers can interpret results consistently.

Results

- Alignment of text with tables/figures: Ensure key claims in the text are supported by tables/figures. For example, the statement “Cluster 1 (those who refuse vaccines) exhibited the lowest uptake across all vaccines …” should be accompanied by a table or figure showing uptake by cluster with appropriate statistics.

- Figure clarity (Figure 3): Improve the figure’s interpretability by enhancing labels Y axis. Also, I do not quit understand the interpretation of this figure.

Writing and organization

- Conciseness: The manuscript would benefit from substantial tightening for length and clarity.

- Logical flow: Present Methods and Results in a stepwise manner aligned with measurement variables.

- Report reliability statistics (e.g., Cronbach’s alpha) in the Results rather than in the Methods.

I hope these comments are helpful in revising the manuscript.

Reviewers' comments:

Reviewer's Responses to Questions

**Comments to the Author**

1. Is the manuscript technically sound, and do the data support the conclusions?

Reviewer #1: Yes

Reviewer #2: Yes

Reviewer #3: Partly

2. Has the statistical analysis been performed appropriately and rigorously?

Reviewer #1: Yes

Reviewer #2: Yes

Reviewer #3: Yes

3. Have the authors made all data underlying the findings in their manuscript fully available?

Reviewer #1: Yes

Reviewer #2: Yes

Reviewer #3: Yes

4. Is the manuscript presented in an intelligible fashion and written in standard English?

Reviewer #1: Yes

Reviewer #2: Yes

Reviewer #3: No

Reviewer #1: Awad et al explore the social determinants influencing childhood vaccine hesitancy among a sample of Lebanese parents, focusing on how individuals are born, grow, live, and work.

1) Why did the authors choose the age group <=3 years of age? Why not broaden the age group?

2) How were the parents selected?

3) Discussion-how do these results compare to other countries and their beliefs? This is important to generalize the results to other countries.

4) Discussion-was there a factor of the distrust that developed during COVID influence some of these beliefs?

5) The results reveal that many parents are dependent on social media for their medical information-please expand on this as this is concerning and how can this be changed?

Reviewer #2: ABSTRACT

1) There was mentioned about involving questionnaire on 'political and economic instability', but not mentioned in the Methods or any other sections of the main text.

METHODS

1) in the sentence "I refuse to give the vaccine to their child", should the word "their" actually be replaced with "my"?

2) Give the full term for SDOH as it is first mentioned in the main text, besides in the abstract.

3) It was mentioned that the SES scale was used to capture respondent financial strains, DMS was to evaluate patients' self-reported experiences of discrimination in healthcare environments, DSES was used to examine parents' religious believes. But the purpose of VHL scale was not stated. Please provide at least a sentence that clearly mention the use of VHL scale.

4) It was stated that vaccine hesitancy scale was used as the dependent variable. But such scale was not described in the method. Please provide methods in determining the dependent variable. If the vaccine hesitancy scale was employed as claimed in the paper, clearly describe the scale like how other scales were described. Also, include explanation on the outcomes of vaccine hesitancy. Are the outcomes in binary outcomes since logistic regression was conducted? Or the outcomes are continuous, like the higher the score, the higher the hesitancy? If they are continuous, shouldn't Linear regression be performed rather than logistic regression?

RESULTS

1) What statistical test was used for results in Table 4? Perhaps can state in the table's footnote.

2) Linear regression was used in Table 6, but the method was not mentioned in the Methods section, instead Logistic regression was mentioned.

3) Why other factors like sociodemographic factors were not included in the analysis of Linear regression in Table 6? Is there any specific reason?

Reviewer #3: Thank you for the opportunity to review this manuscript.

I appreciate the authors’ contribution to the vaccine hesitancy literature. However, several issues should be addressed to strengthen the paper.

Methods

- Questionnaire provenance and content: Please clarify whether the questionnaire was newly developed or adapted from existing instruments. Provide citations and brief descriptions for each scale (e.g., SES, DMS, VHL), including example items or domains, scoring, and interpretation. Define all abbreviations at first use.

- Factor-analytic approach: Clearly describe the factor methods used, including whether the analysis was exploratory or confirmatory. Specify the factor model, provide (or reference) a path/construct diagram, and report key details (extraction and rotation methods, factor retention criteria, factor loadings, cross-loadings, and—if CFA—model fit indices such as CFI/TLI/RMSEA/SRMR).

- Variable definitions and cut-points: Define all derived variables and thresholds (e.g., “high/low” discrimination scale, vaccine health literacy). Explain how cut-points were chosen (prior literature vs data-driven) and ensure directionality is clear so readers can interpret results consistently.

Results

- Alignment of text with tables/figures: Ensure key claims in the text are supported by tables/figures. For example, the statement “Cluster 1 (those who refuse vaccines) exhibited the lowest uptake across all vaccines …” should be accompanied by a table or figure showing uptake by cluster with appropriate statistics.

- Figure clarity (Figure 3): Improve the figure’s interpretability by enhancing labels Y axis. Also, I do not quit understand the interpretation of this figure.

Writing and organization

- Conciseness: The manuscript would benefit from substantial tightening for length and clarity.

- Logical flow: Present Methods and Results in a stepwise manner aligned with measurement variables.

- Report reliability statistics (e.g., Cronbach’s alpha) in the Results rather than in the Methods.

I hope these comments are helpful in revising the manuscript.

**Do you want your identity to be public for this peer review?** For information about this choice, including consent withdrawal, please see our For information about this choice, including consent withdrawal, please see our Privacy Policy .

Reviewer #1: No

Reviewer #2: No

Reviewer #3: No

While revising your submission, please upload your figure files to the Preflight Analysis and Conversion Engine (PACE) digital diagnostic tool, https://pacev2.apexcovantage.com/ . PACE helps ensure that figures meet PLOS requirements. To use PACE, you must first register as a user. Registration is free. Then, login and navigate to the UPLOAD tab, where you will find detailed instructions on how to use the tool. If you encounter any issues or have any questions when using PACE, please email PLOS at . PACE helps ensure that figures meet PLOS requirements. To use PACE, you must first register as a user. Registration is free. Then, login and navigate to the UPLOAD tab, where you will find detailed instructions on how to use the tool. If you encounter any issues or have any questions when using PACE, please email PLOS at figures@plos.org . Please note that Supporting Information files do not need this step.. Please note that Supporting Information files do not need this step.

---

## [Author Response · Author response to Decision Letter 1]

22 Feb 2026

Reviewer #2: Minor revision

ABSTRACT

1) There was mentioned about involving questionnaire on 'political and economic instability’ but not mentioned in the Methods or any other sections of the main text.

Deleted

METHODS

1) in the sentence "I refuse to give the vaccine to their child", should the word "their" actually be replaced with "my"?

Replaced

2) Give the full term for SDOH as it is first mentioned in the main text, besides in the abstract.

Done

3) It was mentioned that the SES scale was used to capture respondent financial strains, DMS was to evaluate patients' self-reported experiences of discrimination in healthcare environments, DSES was used to examine parents' religious beliefs. But the purpose of VHL scale was not stated. Please provide at least a sentence that clearly mention the use of VHL scale.

The Vaccine Health Literacy (VHL) Scale was used to assess parents' ability to obtain, understand, and use vaccine-related information to make informed vaccination decisions for their children. The scale consisted of 14 items divided into three subscales: functional literacy (understanding basic vaccine information), communicative literacy (ability to extract and apply vaccine information), and critical literacy (ability to critically analyze vaccine information). Responses were assessed using a 4-point Likert scale (1 = Never, 2 = Rarely, 3 = Sometimes, 4 = Often), with total scores ranging from 14 to 56. Higher scores indicated better vaccine health literacy. The scale demonstrated good internal consistency (Cronbach's alpha = 0.819) [33,34].

4) It was stated that vaccine hesitancy scale was used as the dependent variable. But such scale was not described in the method. Please provide methods in determining the dependent variable. If the vaccine hesitancy scale was employed as claimed in the paper, clearly describe the scale like how other scales were described. Also, include explanation on the outcomes of vaccine hesitancy. Are the outcomes in binary outcomes since logistic regression was conducted? Or the outcomes are continuous, like the higher the score, the higher the hesitancy? If they are continuous, shouldn't Linear regression be performed rather than logistic regression?

Indeed it is linear regression, not logistic regression.

The second section assessed vaccine hesitancy using a 23-item scale. Vaccine hesitancy was defined for participants as "a delay in acceptance or refusal of vaccines despite the availability of vaccine services." The scale measured two dimensions: vaccine hesitancy (16 items) and vaccine acceptance/trust (7 items). Respondents rated their level of agreement with statements concerning vaccination of children three years old and below using a 5-point Likert scale ranging from 1 (Strongly disagree) to 5 (Strongly agree). The vaccine hesitancy scale score was treated as a continuous variable (range: 23-115) in all analyses. Higher scores on the hesitancy subscale indicate greater vaccine hesitancy, while higher scores on the acceptance/trust subscale indicate more positive attitudes toward vaccination. The complete list of items and their factor loadings is presented in Table 1.

RESULTS

1) What statistical test was used for results in Table 4? Perhaps can state in the table's footnote.

Added it to footnote: Independent samples t-test for continuous variables; Chi-square test for categorical variables.

2) Linear regression was used in Table 6, but the method was not mentioned in the Methods section, instead Logistic regression was mentioned.

Indeed, corrected.

3) Why other factors like sociodemographic factors were not included in the analysis of Linear regression in Table 6? Is there any specific reason?

We considered only variable in the bivariate analysis with p< 0.05

---

## [Editor Report · Decision Letter 1]

3 Mar 2026

Social determinants of vaccine hesitancy among the Lebanese parents: A Cross-sectional Study

PONE-D-25-29851R1

Dear Dr. Awad,

We’re pleased to inform you that your manuscript has been judged scientifically suitable for publication and will be formally accepted for publication once it meets all outstanding technical requirements.

We are asking you to specifically address the comments of Reviewer 3 as soon as possible.

Kind regards,

Vladimir Berthaud

Academic Editor

PLOS One

Additional Editor Comments (optional):

Dear Dr Awad

Thank you for responding to the comments of Reviewer 1 and 2.

We are ready to accept your manuscript for publication, but in order to advance your manuscript, please address the comments of Reviewer 3.

We expect your response within two weeks.

We appreciate your diligence.

Dr. Vladimir Berthaud

Academic Editor
---

## [Editor Report · Acceptance letter]

PONE-D-25-29851R1

PLOS One

Dear Dr. Awad,

I'm pleased to inform you that your manuscript has been deemed suitable for publication in PLOS One. Congratulations! Your manuscript is now being handed over to our production team.

Kind regards,

on behalf of

Dr. Vladimir Berthaud

Academic Editor

PLOS One